# Group 1 mGluR stimulation rescues APOE4-mediated translation defects in neurons

Bindushree K Radhakrishna[1,2,*], Ahamed P Kaladiyil[1,*], Anushree Chakraborty[3], Vini Gautam[3], Ravi S Muddashetty[1]

**The E4 isoform of apolipoprotein (APOE4) is the most recognized risk factor for Alzheimer's disease, implicated in early neurodegeneration and impaired synaptic plasticity. In neurons, exposure to APOE4 disrupts basal and NMDAR-mediated calcium signaling, further disrupting protein synthesis response. Group 1 mGluRs, a major class of glutamate receptors, also play a critical role in synaptic plasticity through activity-dependent protein synthesis. In this study, we examine neuronal protein synthesis response to mGluR stimulation in the background of APOE4 treatment. In DIV15 primary cortical neurons from Sprague-Dawley rat embryos, exposure to APOE4 induces inhibition of protein synthesis, which is rescued by stimulation of mGluRs for 5 min. mGluR stimulation also rescued the APOE4-induced reduction in synaptic activity as measured by the multielectrode array. This mGluR-mediated rescue is driven by phosphorylation of RPS6, downstream of the mammalian target of rapamycin (mTOR) pathway as it is abolished by rapamycin treatment. This p-RPS6–driven rescue is independent of calcium-mediated translation inhibition induced by APOE4, demonstrating a specific and independent role of mTORC1 activity in maintaining mGluRs' translation capacity under APOE4 exposure. The potential of mGluR-mediated response to compensate for the effect of APOE4 suggests a dynamic mechanism for the induction of plasticity in human APOE4 carriers. This study provokes a critical need to explore the altered synaptic dynamics in the presence of APOE4 and its impact on cognition.**

## Introduction

The E4 allele of apolipoprotein E (APOE4) is the most extensively studied genetic risk factor for Alzheimer's disease (AD). Beyond its role in AD, APOE4 is also associated with other neurodegenerative disorders, where its presence is linked to increased risk of dementia in a dose-dependent manner (Frisoni et al, 1995; Pankratz et al, 2006). The best characterized effects of having the E4 allele are an increased propensity to form Aβ aggregates in the brain and an increase in tau phosphorylation in neurons, associated with synapse loss and neurodegeneration (Liraz et al, 2013; Zhao et al, 2020).

Defective synaptic activity and synapse loss are a major correlate of cognitive impairment (Terry et al, 1991; Silva, 2003). APOE4 is known to impair several aspects of synaptic functioning. APOE4-expressing mice exhibit reduced spontaneous excitatory synaptic transmission and attenuated induction of long-term potentiation (LTP) in the hippocampus (Trommer et al, 2004; Wang et al, 2005; Qiao et al, 2014). Although APOE4 impairs excitatory synaptic transmission and LTP in murine and in vitro models, there is a lack of understanding of the molecular mechanisms of synaptic plasticity in the background in APOE4.

At glutamatergic synapses, synaptic plasticity is primarily mediated by NMDARs and mGluRs (Zho et al, 2002; Rebola et al, 2010). The interplay between these receptors is crucial for regulating neuronal excitability and plasticity. The maintenance of neuronal protein content and translation rate is essential for maintaining synaptic health, as it directly influences the availability of key synaptic proteins required for synaptic function and plasticity (Cajigas et al, 2010). NMDARs and mGluRs display distinct activity–mediated protein synthesis responses, implying different roles in plasticity (Ghosh Dastidar et al, 2020). Although the NMDAR-mediated translation response is primarily driven by calcium signaling, the mGluR-mediated translation response is only partially controlled by calcium (Ramakrishna et al, 2024). mGluRs play a modulatory role in enhancing NMDAR-dependent plasticity by reducing the signal-to-noise ratio. One mechanism through which this occurs is mGluR1-mediated phosphorylation of the NR2A/B subunits of NMDARs, which strengthens LTP induction (Heidinger et al, 2002).

Because plasticity mechanisms are heavily reliant on activity-mediated protein synthesis, it is crucial to study the receptor-induced protein synthesis responses affected by APOE4 (Sutton & Schuman, 2005). Exposure to APOE4 is reported to cause transcriptional activation of multiple mRNAs within 48 h and increased translation of synaptic proteins such as PSD95 and GluA1 as early as 20 min on treatment (Huang et al, 2019; Wang et al, 2022). APOE4 was shown to affect global protein synthesis through the activation of NMDA receptors. APOE4-mediated calcium influx

[1]Centre for Brain Research, Indian Institute of Science, Bangalore, India   [2]Manipal Academy of Higher Education, Manipal, India   [3]Centre for Nano Science and Engineering, Indian Institute of Science, Bangalore, India

Correspondence: ravimshetty@cbr-iisc.ac.in
*Bindushree K Radhakrishna and Ahamed P Kaladiyil contributed equally to this work

resulted in increased phosphorylation of the eukaryotic elongation factor 2 (eEF2), leading to a persistent suppression of translation elongation, a feature not observed on APOE3 treatment. The increased calcium influx induced by APOE4 also abrogated NMDAR-mediated protein translation response in neurons (Ramakrishna et al, 2021). However, the implications of APOE4's persistent decrease in translation on cellular functioning and synaptic plasticity are not well understood. APOE4, therefore, has a very robust impact on synaptic functions in in vitro and animal models; however, its significant impact on cognition is observed primarily in the elderly human population.

Given that APOE4 impairs NMDAR-mediated calcium signaling and protein synthesis response, it is important to explore whether mGluRs could have compensatory roles in the background of APOE4. Plasticity mechanisms induced by mGluRs are dependent on receptor activation–induced increase in protein synthesis (Huber et al, 2000). This protein synthesis response in neurons is primarily dependent on the PI3K-AKT-mTOR–mediated increase in translation initiation (Hou & Klann, 2004). To understand any effects of APOE4 on mGluR-induced plasticity mechanisms or to identify any role of mGluR in rescuing synaptic deficits, it is essential to investigate mGluR-dependent translation response in the background of APOE4. In the current study, we show that stimulation of group I mGluRs in the presence of APOE4 rescues the protein synthesis inhibition and this process is mediated through an increase in ribosomal protein S6 (RPS6) phosphorylation, downstream of the mTOR signaling pathway.

# Results

### Group 1 mGluR stimulation rescues APOE4-mediated reduction in neuronal de novo protein synthesis and synaptic activity defect

Exposure to APOE4 significantly reduces the protein synthesis in neurons as compared to APOE3. This protein synthesis inhibition is due to aberrantly increased calcium influx, which also abrogates NMDAR-mediated translation response in these neurons (Ramakrishna et al, 2021). Because metabotropic glutamate receptors (mGluRs) are the other major class of glutamate receptors that elicit distinct activity–mediated translational response in neurons (Muddashetty et al, 2007; Ghosh Dastidar et al, 2020), we studied the effect of group I mGluR stimulation on APOE4 exposure. DIV15 rat primary cortical neurons were treated with human recombinant protein APOE4 (10 nM) for 20 min. Using 50 $\mu$M of (S)-3,5-dihydroxyphenylglycine (DHPG), group 1 mGluRs were stimulated during the last 5 min of APOE4 exposure (Fig 1A). The ratio of FUNCAT signal to Tuj1 ($\beta$-III-tubulin) intensity was obtained from the selected region of interest (ROI) and normalized to the untreated condition to measure the relative change in de novo protein synthesis.

In consensus with previous findings, an increase in neuronal protein synthesis was observed when treated with DHPG for 5 min (Huber et al, 2000; Muddashetty et al, 2007; Ghosh Dastidar et al, 2020), and a decrease in neuronal protein synthesis on 20 min of APOE4 treatment (Ramakrishna et al, 2021). Interestingly, 5 min of

DHPG treatment (mGluR stimulation) in APOE4-treated neurons markedly increased protein levels as compared to only APOE4-treated neurons with no significant change as compared to the untreated condition (Fig 1B and C). Whole neuron intensity was measured from the ROI selected from neurons and its projections visibly expressing the neuronal marker, Tuj1.

Next, to focus on the protein synthesis from the dendritic area, the signal was blocked from the cell body of the neuron, and FUNCAT and Tuj1 intensities were measured exclusively from the dendrites (Fig 1B; bottom). FUNCAT intensity in the dendrites showed similar responses as observed in whole neurons, where DHPG treatment (mGluR stimulation) led to increased FUNCAT intensity, and APOE4 treatment led to a decreased FUNCAT intensity. As observed before (Fig 1C), 5 min of mGluR stimulation in the presence of APOE4 led to an increase in protein levels as compared to just APOE4 treatment rescuing de novo protein synthesis to basal levels (Fig 1B and D). No significant difference was observed in Tuj1 intensity (normalizing control for FUNCAT signal) on APOE4 or DHPG treatments in the whole neuron or dendrite measurements. These results indicate that the inhibition of protein synthesis caused by exposure of neurons to APOE4 is rescued to basal levels by 5 min of group 1 mGluR stimulation.

To test the possibility that APOE4 exposure can affect the mRNA stability contributing to APOE4-mediated protein synthesis reduction, we chose to investigate the levels of mRNA candidates such as Camk2α—a postsynaptic enzyme; Homer1—a postsynaptic scaffolding protein; and α-Tubulin—a cytoskeletal protein, and the levels of the PABP1 protein, which is essential for mRNA stability. APOE4 treatment for 20 min did not alter levels of PABP1 and the mRNA levels of Camk2α, Homer1, and α-Tubulin (Fig S1A and B). These results indicate that APOE4 treatment does not affect mRNA stability, suggesting that the decreased FUNCAT intensity is due to protein synthesis inhibition.

APOE4 has been consistently linked to impaired synaptic plasticity and reduced neuronal activity (Chen et al, 2010; Har-Paz et al, 2021), whereas mGluR stimulation has been shown to enhance evoked synaptic activity responses (Morris et al, 1999). To investigate how APOE4 influences synaptic function, we assessed neuronal ability to undergo potentiation using multielectrode array (MEA) recordings in DIV15 rat primary cortical neurons (Fig 1E). In untreated neurons, high-frequency stimulation (HFS) produced a robust increase in mean spike rate, reflecting a typical evoked response (Figs 1F and S1D). In contrast, neurons treated with APOE4 for 20 min followed by the first HFS and the same neurons subjected to the second HFS after 24 h failed to exhibit evoked response indicating impaired synaptic potentiation (Figs 1G and H and S1E and F). Notably, mGluR stimulation using DHPG rescued this defect: APOE4-exposed neurons treated with DHPG (5 min at the end of the 20-min treatment) displayed a significant increase in spike rate post the first HFS, similar to untreated neurons (Figs 1G and S1E). A similar rescue effect was observed post the second HFS administered on the same neurons (APOE4- and DHPG-treated) after 24 h (Figs 1H and S1F). These findings demonstrate that mGluR stimulation not only counteracts APOE4-mediated inhibition of protein synthesis but also restores synaptic activity and plasticity.

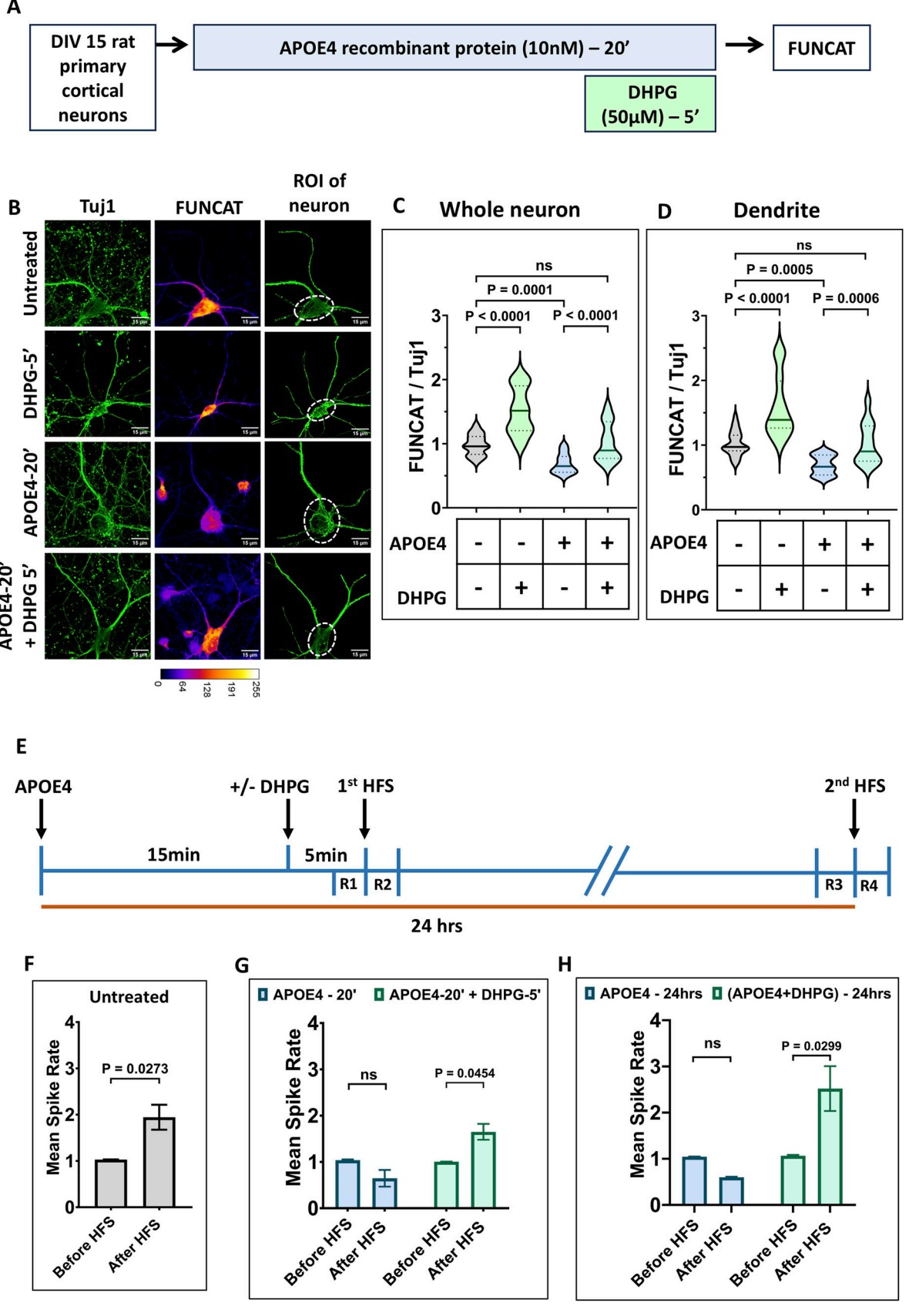

### mGluR stimulation leads to increased RPS6 phosphorylation in the presence of APOE4, whereas the p-eEF2 level remains unaltered

To delineate the signaling components that lead to mGluR-mediated protein synthesis rescue, the phosphorylation of RPS6, a well-known downstream response of mGluR stimulation (Antion et al, 2008), was studied on APOE4 exposure. eEF2 phosphorylation was also measured as a readout of translation inhibition elicited by APOE4 (Ramakrishna et al, 2021). Neurons were immunostained for p-RPS6 or p-eEF2 and Tuj1 (for normalization) after APOE4 and DHPG treatment (mGluR stimulation) (Fig S2A). p-RPS6 and p-eEF2 levels were also measured using immunoblotting under the same treatment conditions (Fig 2A).

In neurons, 5 min of mGluR stimulation using DHPG led to an increase in phosphorylation of RPS6. APOE4 treatment for 20 min led to no significant changes in p-RPS6 levels. 5 min of DHPG treatment (mGluR stimulation) in the background of APOE4 resulted in a significant increase in p-RPS6, implying that a mGluR-mediated increase in p-RPS6 remains unaffected in the background of APOE4 (Fig 2B–D). A mGluR-mediated increase in p-RPS6 was also observed in dendrites, which was unaffected by the presence of APOE4 (Fig S2B and C).

As reported previously, 20 min of APOE4 exposure evoked an increase in phosphorylation of eEF2, which led to translation inhibition (Ramakrishna et al, 2021). 5 min of DHPG treatment (mGluR stimulation) alone elicited no significant change in the p-eEF2 signal. In the background of APOE4, mGluR stimulation did not have an effect on p-eEF2 levels (Fig 2E–G). Dendritic measurements also led to a similar observation, where APOE4 induced an increase in p-eEF2 levels and mGluR stimulation did not change the p-eEF2 levels in the background of APOE4 (Fig S2D and E).

In summary, mGluR stimulation increases protein synthesis through an increased rate of translation initiation mediated by phosphorylation of RPS6 (Ramakrishna et al, 2024). This response remained unaffected in neurons even in the presence of APOE4. The translation inhibition induced by APOE4, mediated by increased p-eEF2 levels, remained unaffected by mGluR stimulation. Thus, mGluR-mediated translation rescue primarily occurs by up-regulating translation initiation to increase overall protein synthesis.

### Activation of the mTOR pathway drives mGluR-mediated protein synthesis response in the background of APOE4

To further establish the role of increased p-RPS6 in driving mGluR-mediated translation rescue, rapamycin was used to inhibit an upstream activator of RPS6 phosphorylation. Rapamycin has previously been shown to specifically inhibit the activity of mTORC1, the upstream activator of S6K, the primary kinase involved in phosphorylation of RPS6 (Holz & Blenis, 2005). Thus, using rapamycin, we studied the contribution of mTORC1 activity in mGluR-mediated protein synthesis rescue of APOE4-induced translation inhibition.

To avoid the interference between rapamycin-induced mTORC1 inhibition and APOE4's translation inhibition response, the duration of rapamycin treatment was limited to the duration of mGluR stimulation. To test the effect of rapamycin on abolishing mGluR-mediated FUNCAT and p-RPS6 response, neurons were treated with rapamycin (500 nM) along with DHPG for 5 min (Fig 3A). The treatment of neurons with rapamycin (500 nM) completely inhibited mGluR-mediated increase in FUNCAT (Fig 3E and F) and p-RPS6 signal (Fig 4E and F). Thus, the simultaneous addition of 500 nM of rapamycin with DHPG was chosen for inhibiting mGluR response in this context.

To further verify rapamycin's effect on APOE4, we measured changes in FUNCAT levels, which showed that 5 min of rapamycin treatment to APOE4-exposed neurons did not affect APOE4's FUNCAT response. Furthermore, 5 min of rapamycin treatment alone was found to reduce FUNCAT signal as expected and in line with other studies (Fig 3G and H).

After this, rapamycin and DHPG treatments were performed in the background of APOE4 to study FUNCAT and p-RPS6 response (Figs 3A and 4A). On rapamycin treatment, FUNCAT measurements showed the abolishment of mGluR-mediated rescue of APOE4's translation defect (Fig 3B and C). Rapamycin-mediated inhibition of mTORC1 activity in the background of APOE4 completely suppressed mGluR-mediated increase in p-RPS6 (Fig 4B and C). Similar observations were made on measuring FUNCAT and p-RPS6 levels from dendrites (Figs 3B and D and 4B and D). These results indicate that the mGluR-dependent protein synthesis rescue of APOE4-induced translation inhibition is dependent on mTORC1 activity.

**Figure 1. mGluR stimulation rescues APOE4-mediated protein synthesis inhibition and synaptic activity defect.**
**(A)** Schematic showing the workflow of the experiment. DIV15 rat primary cortical neurons were treated with 10 nM APOE4 recombinant protein for 20 min. The neurons were stimulated with 50 $\mu$M DHPG in the last 5 min. FUNCAT was performed to quantify de novo protein synthesis. **(B)** Representative images of DIV15 primary cortical neurons indicating Tuj1, and FUNCAT intensities on APOE4 treatment with or without DHPG. The Tuj1 channel was used to select the region of interest (ROI) of the neuron for analysis. Scale bar—15 $\mu$m. **(C)** Quantification of the FUNCAT intensity of the whole neuron normalized to Tuj1 intensity of the whole neuron (ROI of the neuron indicated) on APOE4 treatment with or without DHPG. N = 27–29 neurons from three independent experiments, one-way ANOVA ($P < 0.0001$) followed by Tukey's multiple comparison test. **(D)** Quantification of the FUNCAT intensity in dendrites normalized to Tuj1 intensity in dendrites (area marked by a dotted line in the ROI of the neuron was excluded for dendrite analysis) on APOE4 treatment with or without DHPG. N = 27–29 neurons from three independent experiments, one-way ANOVA ($P < 0.0001$) followed by Tukey's multiple comparison test. **(E)** Schematic showing the workflow of the multielectrode array experiment. DIV15 rat primary cortical neurons were treated with 10 nM APOE4 recombinant protein for 20 min with or without 50 $\mu$M DHPG treatment in the last 5 min. The neurons were then subjected to high-frequency stimulation (first HFS), and the spike rate of the neurons was recorded before HFS (R1—2 min) and after HFS (R2—2 min). The second HFS was carried out on the same neurons after 24 h (DIV16) of APOE4 treatment with or without DHPG, and the spike rate of neurons was recorded before HFS (R3—2 min) and after HFS (R4—2 min). **(F)** Bar graph showing the mean spike rate of the untreated neurons after HFS was normalized to mean spike rate before HFS. Data are represented as the mean ± SEM, N = 3, paired t test. **(G)** Bar graph showing the mean spike rate of the neurons after the first HFS (APOE4-20′ ± DHPG-5′) normalized to the mean spike rate before HFS (R2/R1). Data are represented as the mean ± SEM, N = 3, two-way ANOVA followed by Sidak's multiple comparison test. **(H)** Bar graph showing the mean spike rate of the neurons after the second HFS (APOE4 ± DHPG 24 h) normalized to the mean spike rate before HFS (R4/R3). Data are represented as the mean ± SEM, N = 3, two-way ANOVA followed by Sidak's multiple comparison test.
Source data are available for this figure.

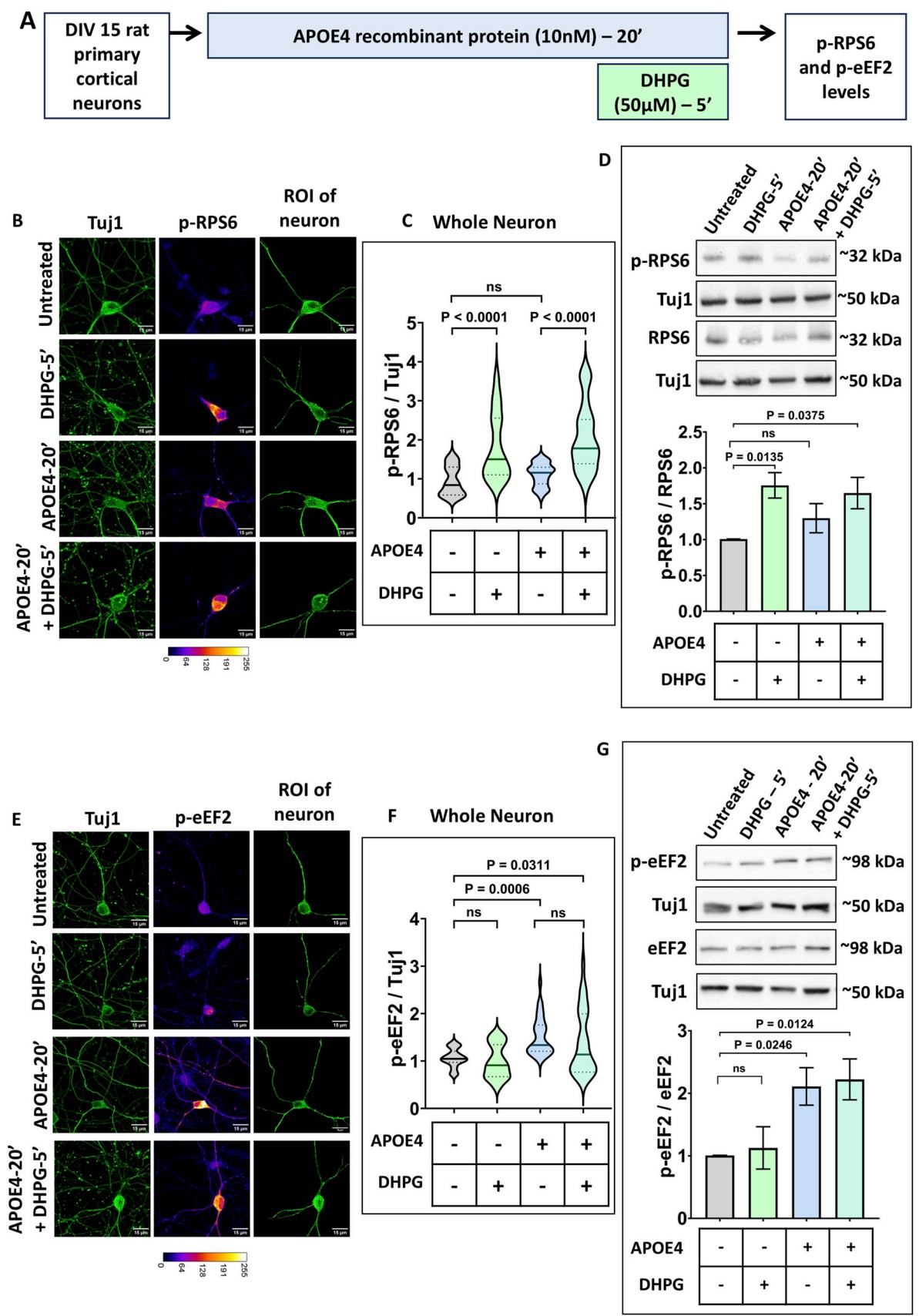

To rule out any effects of rapamycin on APOE4-mediated translation inhibition, p-eEF2 levels were measured on the simultaneous addition of rapamycin and DHPG on APOE4 pretreatment (Fig 4A). An APOE4-induced increase in p-eEF2 levels remained unchanged by the addition of rapamycin (Fig 4G–I). Thus, rapamycin (mTORC1 inhibition) did not elicit any change in APOE4's increased p-eEF2 response, implying no role of mTORC1 and its downstream signaling in APOE4-induced translation inhibition (Fig 5).

# Discussion

Our previous work demonstrated that APOE4 perturbs neuronal protein synthesis by excessive calcium influx through NMDARs and L-type voltage-gated calcium channels (L-VGCCs), leading to sustained phosphorylation of eEF2 and consequent suppression of translation elongation (Ramakrishna et al, 2024). This disruption was found to not only diminish basal protein synthesis but also attenuated NMDAR-dependent translation response. On the other hand, studies have shown that APOE4 compromises synaptic plasticity by interfering with glutamate receptor trafficking, reducing NMDAR phosphorylation, and impairing downstream effectors such as CaMKIIα and CREB, thereby weakening activity-dependent protein renewal and long-term potentiation (Chen et al, 2010; Qiao et al, 2014; Huang et al, 2019).

Besides NMDARs, group I mGluR activity is essential for the maintenance and induction of plasticity mechanisms. The stimulation of mGluRs leads to a robust translation activation through mTOR-dependent phosphorylation of ribosomal protein S6 (RPS6) (Costa-Mattioli et al, 2009). mGluRs increase translation and preferentially up-regulate proteins such as Arc and MAP1B, which govern receptor trafficking and spine remodeling, enabling specific forms of plasticity (Waung et al, 2008; Chen & Shen, 2013), and modulate the excitability in a broader region, whereas NMDARs regulate synapse-specific activity. These mechanistic differences raised the possibility that mGluR-driven translation might be spared in the context of APOE4, thereby providing an alternative route to sustain proteostasis.

Consistent with this hypothesis, we found that mGluR stimulation restored basal protein synthesis capacity in APOE4-treated neurons. Mechanistically, this rescue was mediated by enhanced phosphorylation of RPS6 through mTOR activation, whereas the APOE4-induced elevation of p-eEF2 remained unaltered. Thus, mGluR signaling appears to restore the translational output by a pathway independent of APOE4-mediated signaling. Notably, pharmacological inhibition of mTOR with rapamycin abolished the rescue effect of mGluR stimulation, underscoring an essential role of mTOR in this. This context-dependent requirement for mTOR is particularly striking, given that rapamycin is widely exploited in cancer research as a potential therapeutic molecule, yet here, its blockade prevents the restoration of protein synthesis and synaptic plasticity in the presence of APOE4 (Fig 5).

Beyond translation, we show that mGluR activation reinstated HFS-induced LTP in APOE4-exposed neurons, a form of plasticity otherwise abolished in this background. An APOE4-induced defect in LTP induction is widely reported in previous studies linking APOE4 to impaired CaMKIIα and CREB activation (Chen et al, 2010; Qiao et al, 2014). Mechanistically, mGluR signaling likely bypasses the defective NMDAR–calcium axis by rerouting activity-dependent cascades through alternative kinase/phosphatase pathways. Moreover, mGluR5 is known to interact with NMDARs via Homer–Shank scaffolds and PKC signaling (Collett & Collingridge, 2004; Chen et al, 2011; Sylantyev et al, 2013), raising the possibility that mGluR stimulation not only circumvents but also partially resensitizes NMDAR function under APOE4 conditions. It is also conceivable that mGluR-mediated rescue occurs by affecting the dynamics of postsynaptic receptor endocytosis and recycling or altering the phosphorylation statuses of glutamate receptor subunits (Choe et al, 2006; Chen et al, 2011). An independent study shows that restoring vesicular dynamics in APOE4-expressing mice can restore LTP induction (Pohlkamp et al, 2021), suggesting that the LTP defect arises from altered expression and activity of glutamate receptors at the postsynaptic membrane. However, it is also to be noted that mGluR-mediated receptor endocytosis and recycling processes are dependent on de novo protein synthesis (Snyder et al, 2001), further highlighting the importance of activity-mediated protein synthesis.

Synaptic plasticity is an emergent property of diverse cellular processes, among which protein synthesis constitutes a major determinant. Although our findings establish that mGluR stimulation rescues both protein synthesis and evoked activity, the molecular underpinnings of this effect remain to be fully delineated. Future work should address whether mGluR-mediated

**Figure 2. mGluR stimulation up-regulates p-RPS6 in the presence of APOE4, whereas p-eEF2 response is unaltered.**
**(A)** Schematic showing the workflow of the experiment. DIV15 rat primary cortical neurons were treated with 10 nM APOE4 recombinant protein for 20 min. The neurons were stimulated with 50 μM DHPG in the last 5 min. p-RPS6 and p-eEF2 levels were quantified by immunostaining and immunoblotting. **(B)** Representative images of DIV15 primary cortical neurons indicating Tuj1 and p-RPS6 intensities on APOE4 treatment with or without DHPG. The Tuj1 channel was used to select the region of interest (ROI) of the neuron for analysis. Scale bar—15 μm. The same images are represented in Fig S2B. **(C)** Quantification of the p-RPS6 intensity in the whole neuron normalized to Tuj1 intensity in the whole neuron (ROI of the neuron indicated) on APOE4 treatment with or without DHPG. N = 29–31 neurons from three independent experiments, one-way ANOVA (P < 0.0001) followed by Tukey's multiple comparison test. **(D) Top:** Representative immunoblots indicating p-RPS6, RPS6, and Tuj1 levels in DIV15 primary cortical neurons on APOE4 treatment with or without DHPG. **Bottom:** quantification of the p-RPS6-to-RPS6 ratio on APOE4 treatment with or without DHPG. Data are represented as the mean ± SEM. N = 8, one-way ANOVA followed by Dunnett's multiple comparison test. **(E)** Representative images of DIV15 primary cortical neurons indicating Tuj1 and p-eEF2 intensities on APOE4 treatment with or without DHPG. The Tuj1 channel was used to select the ROI of the neuron for analysis. Scale bar—15 μm. The same images are represented in Fig S2D. **(F)** Quantification of the p-eEF2 intensity in the whole neuron normalized to Tuj1 intensity in dendrites (ROI of the neuron indicated) on APOE4 treatment with or without DHPG. N = 30–37 neurons from three independent experiments, one-way ANOVA (P < 0.0001) followed by Tukey's multiple comparison test. **(G) Top:** representative immunoblots indicating p-eEF2, eEF2, and Tuj1 levels in DIV15 primary cortical neurons on APOE4 treatment with or without DHPG. **Bottom:** quantification of the p-eEF2-to-eEF2 ratio on APOE4 treatment with or without DHPG. Data are represented as the mean ± SEM. N = 8, one-way ANOVA followed by Dunnett's multiple comparison test.
Source data are available for this figure.

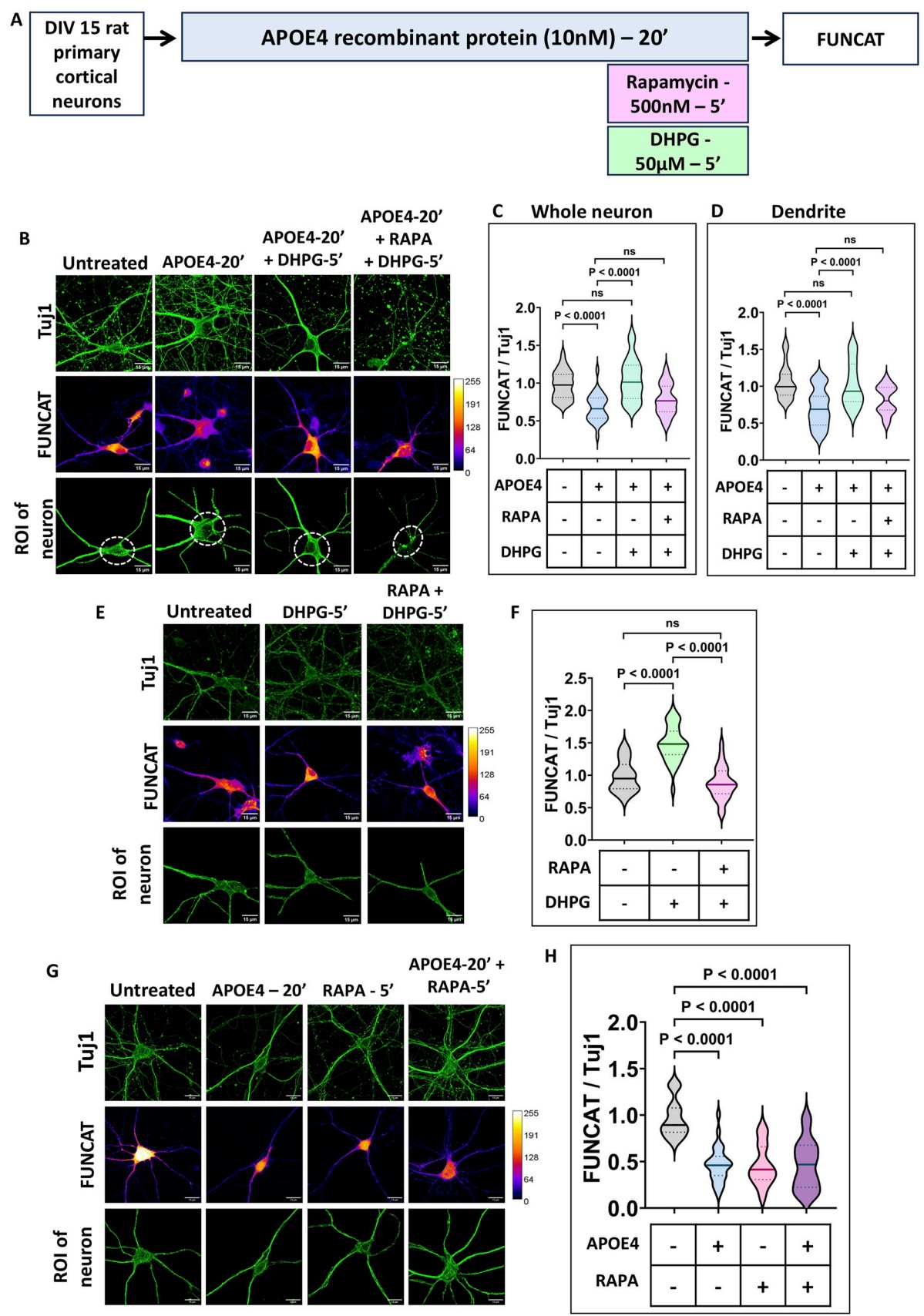

rescue involves modulation of receptor phosphorylation, surface trafficking, or engagement of CREB-dependent transcriptional programs such as BDNF expression.

Taken together, these data provide the first evidence that group I mGluR signaling remains operational in the presence of APOE4 and can restore both translational capacity and synaptic potentiation. By engaging mTOR-dependent initiation, mGluR stimulation offsets APOE4-induced repression of protein synthesis and reinstates activity-driven responses. These findings reveal a previously unrecognized mechanism of synaptic resilience and suggest new therapeutic opportunities aimed at counteracting APOE4-associated cognitive decline.

# Materials and Methods

### Ethics statement

All animal work was conducted in accordance with the institutional guidelines for the care and use of laboratory animals under the approval of the Institutional Animal Ethics Committee (IAEC) and the Institutional Biosafety Committee (IBSC), Centre for Brain Research, Indian Institute of Science Campus, Bangalore, India. The rats used in the study were Sprague-Dawley (SD) rats. The animals were housed and maintained under pathogen-free conditions in a temperature-controlled room on a 12-h light/12-h dark cycle with ad-libitum access to food and water.

### Rat primary neuronal cultures

Primary neuronal cultures were prepared from the cerebral cortices of Sprague-Dawley rat embryos (E18.5), following a protocol previously published by our laboratory (Ghosh Dastidar et al, 2020; Ramakrishna et al, 2024). Briefly, the cortices were trypsinized for 5 min at 37°C using 0.25% trypsin (15050057; Thermo Fisher Scientific). After dissociation, the cells were plated onto cell culture plates containing coverslips at a density of 30,000–35,000 cells/cm$^2$ for imaging-based experiments. Neurons were plated in Minimum Essential Media (MEM, 10095080; Thermo Fisher Scientific) supplemented with 10% FBS to promote attachment. After 3 h,

the MEM was replaced with Neurobasal medium (21103049; Thermo Fisher Scientific), supplemented with 2% B27 (17504044; Thermo Fisher Scientific) and 1X GlutaMAX. The neurons were maintained for 15–17 d at 37°C with 5% CO$_2$, with Neurobasal medium supplementation every 5–6 d. Cell culture plates with coverslips were coated overnight at 37°C with poly-L-lysine (P2636; Sigma-Aldrich) solution at a concentration of 0.1 mg/ml, made in borate buffer (pH 8.5) before neuron plating.

### FUNCAT (fluorescent non-canonical amino acid tagging)

Metabolic labeling with L-azidohomoalanine (AHA) was employed to assess de novo protein synthesis, as described previously by our laboratory (Ghosh Dastidar et al, 2020; Ramakrishna et al, 2024). DIV15 cortical neurons were starved of methionine for 30 min using methionine-free DMEM (21013024; Thermo Fisher Scientific). After starvation, neurons were treated with 1 $\mu$M L-azidohomoalanine (AHA, 1066100; Click Chemistry Tools) for 30 min in methionine-free DMEM. 10 min after AHA incorporation, neurons were treated with 10 nM human APOE4 recombinant protein (350-04; PeproTech) for 20 min. During the last 5 min of APOE4 treatment, mGluR stimulation was induced using 50 $\mu$M DHPG (D3689; Sigma-Aldrich). For experiments involving rapamycin, neurons were cotreated with DHPG and 500 nM rapamycin (53123-88-9; Calbiochem) simultaneously.

After treatments, neurons were fixed with 4% PFA for 15 min. After three washes with 1X PBS, neurons were permeabilized for 10 min with 0.3% Triton X-100 solution in TBS50 (50 mM Tris, 150 mM NaCl, pH 7.6), then blocked for 1 h with a mixture of 2% BSA and 2% FBS in TBS50T (TBS50 with 0.1% Triton X-100). After blocking, neurons were incubated for 3 h with the FUNCAT reaction, following the kit's instructions to tag AHA-incorporated proteins with an alkyne-fluorophore Alexa Fluor 555 through the click reaction (C10269; CLICK-iT Cell Reaction Buffer Kit, Click Chemistry Tools). After three washes with TBS50T, neurons were incubated overnight with Tuj1 antibody at 4°C, followed by a 1-h incubation with a secondary antibody at RT for visualization of $\beta$-III-tubulin (see Table 1 for antibody details).

Coverslips were mounted on glass slides with Mowiol mounting media and imaged using an Olympus FV3000 confocal laser scanning upright microscope with a 60X objective. The pinhole was

**Figure 3. Activation of the mTOR pathway is responsible for mGluR-mediated protein synthesis rescue in the presence of APOE4.**
**(A)** Schematic showing the workflow of the experiment. DIV15 rat primary cortical neurons were treated with 10 nM APOE4 recombinant protein for 20 min. The neurons were stimulated with 50 $\mu$M DHPG in the last 5 min in the presence or absence of rapamycin (500 nM) followed by FUNCAT. **(B)** Representative images of DIV15 primary cortical neurons indicating Tuj1 (top) and FUNCAT intensities (middle) on APOE4 treatment followed by DHPG stimulation with or without rapamycin. The Tuj1 channel was used to select the region of interest (ROI) of the neuron (bottom) for analysis. Scale bar—15 $\mu$m. **(C)** Quantification of the FUNCAT intensity in the whole neuron normalized to Tuj1 intensity in the whole neuron (ROI of the neuron indicated) on APOE4 treatment followed by DHPG stimulation with or without rapamycin. N = 28–33 neurons from three independent experiments, one-way ANOVA ($P < 0.0001$) followed by Tukey's multiple comparison test. **(D)** Quantification of the FUNCAT intensity in dendrites normalized to Tuj1 intensity in dendrites (area marked by a dotted line in the ROI of the neuron was excluded for dendrite analysis) on APOE4 treatment with or without DHPG. N = 28–33 neurons from three independent experiments, one-way ANOVA ($P < 0.0001$) followed by Tukey's multiple comparison test. **(E)** Representative images of DIV15 primary cortical neurons indicating Tuj1 (top) and FUNCAT intensities (middle) on DHPG treatment with or without rapamycin. The Tuj1 channel was used to select the ROI of the neuron (bottom) for analysis. Scale bar—15 $\mu$m. **(F)** Quantification of the FUNCAT intensity in the whole neuron normalized to Tuj1 intensity in the whole neuron (ROI of the neuron indicated) on DHPG treatment with or without rapamycin. N = 27–33 neurons from three independent experiments, one-way ANOVA ($P < 0.0001$) followed by Tukey's multiple comparison test. **(G)** Representative images of DIV15 primary cortical neurons indicating Tuj1 (top) and FUNCAT intensities (middle) on APOE4 treatment with or without rapamycin. The Tuj1 channel was used to select the ROI of the neuron (bottom) for analysis. Scale bar—15 $\mu$m. **(H)** Quantification of the FUNCAT intensity in the whole neuron normalized to Tuj1 intensity in the whole neuron (ROI of the neuron indicated) on APOE4 treatment with or without rapamycin. N = 25–29 neurons from three independent experiments, one-way ANOVA ($P < 0.0001$) followed by Tukey's multiple comparison test.
Source data are available for this figure.

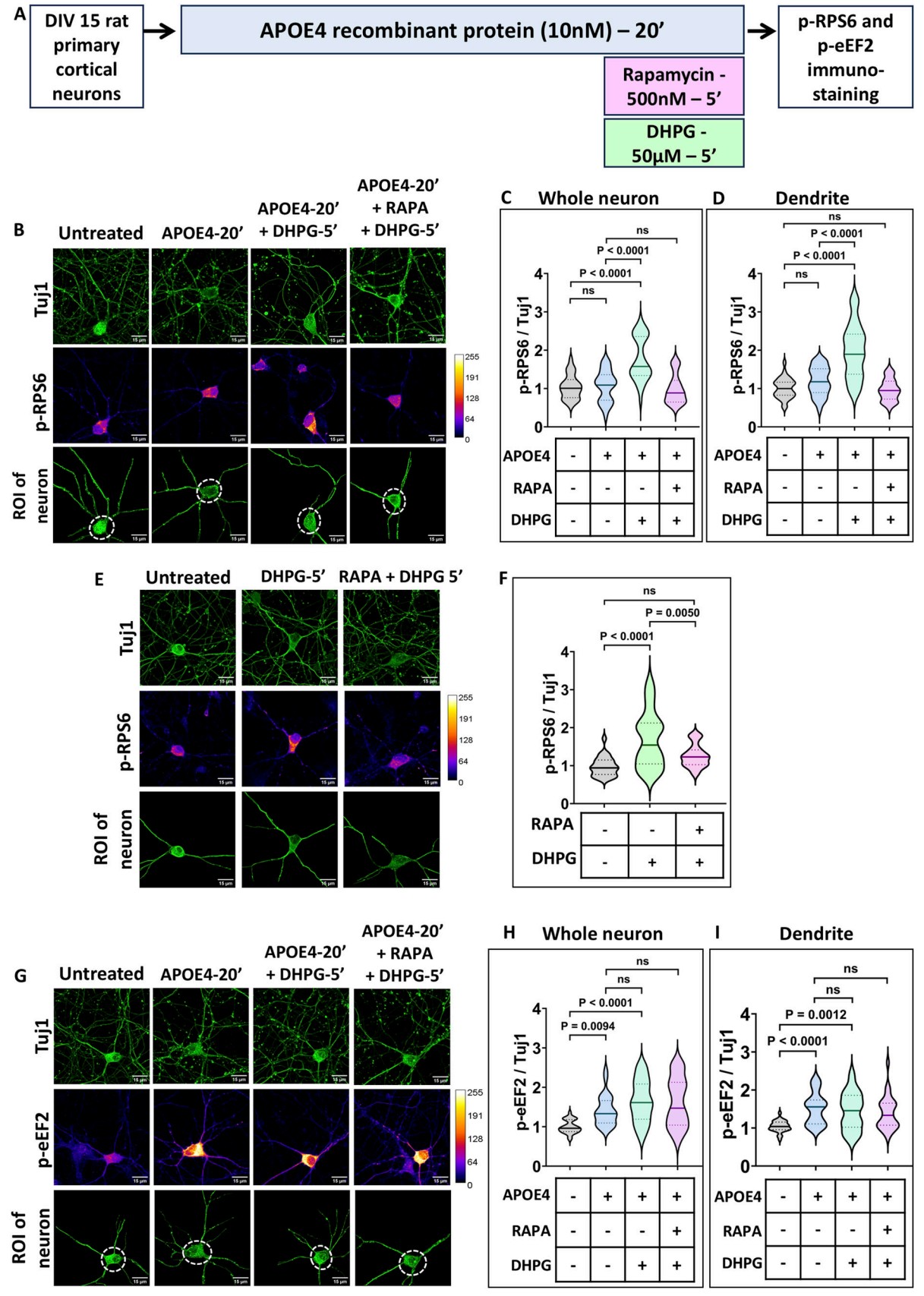

set to 1 Airy Unit, and the optical zoom was set at 2X to meet Nyquist sampling criteria in the XY direction. Z-direction imaging was performed with a 1 $\mu m$ step size, capturing ~10–12 Z-slices to cover planes above and below the focal plane. Image analysis was performed using FIJI software, and the maximum intensity projection of the Z-slices was used for quantification. The Tuj1 channel was used to define the ROI for neurons. For dendritic measurements, the cell body intensity was blocked and measurements were taken from the remainder of the previously selected ROI. The mean fluorescence intensity (total intensity normalized to the area) was measured for both the FUNCAT and Tuj1 channels within the defined ROIs. The mean fluorescence intensity of the FUNCAT channel was normalized to the Tuj1 channel for comparison across treatment conditions.

## Primary neuronal culture on MEA chips

Neuronal culture growth and maintenance were done similar to the procedures followed in Beaudoin et al (2012). Before cell culture, the MEA chips (120MEA200/30iR-Ti; Multichannel Systems) were plasma-treated for 2 mins and then coated with 50 $\mu g/ml$ of poly-L-ornithine (P3655; Sigma-Aldrich) and 5 $\mu g/ml$ of laminin (L2020; Sigma-Aldrich) solution, each having an incubation period of ~12 h in a RH incubator at 37°C, 5% $CO_2$. Rat pups (Wistar, 0–2 d old) were euthanized by decapitation, according to the approved protocols by the Institutional Animal Ethics Committee (IAEC), Indian Institute of Science, Bangalore, India. The cortical tissue was isolated in Hanks' Balanced Salt Solution (14175079; Gibco) with glucose (0.1%), sodium pyruvate (11 mg/ml), and Hepes (10 mM). The tissues were treated with 2.5% trypsin (15090046; Gibco) for 30 mins and then DNase (DN25; Sigma-Aldrich) for 5 mins. The solution was aspirated and replaced by MEM alpha (12561056; Gibco) with 10% FBS, 0.45% glucose, 2 mM glutamine, and 1% penicillin–streptomycin, and homogeneous suspension of cells was obtained upon mechanical dissociation of the tissue. Cell counting was done using a Neubauer hemocytometer chamber using trypan blue dye and plated on MEA chips at a density of 0.75 × $10^3$ cells/$mm^2$ and maintained in an RH incubator at 37°C, 5% $CO_2$ in media

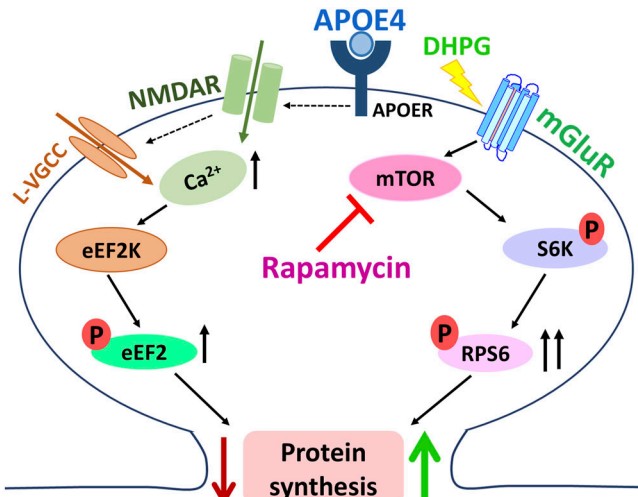

**Figure 5. Model representing the mGluR-mediated rescue of APOE4-induced protein synthesis inhibition.**
APOE4 treatment increases calcium influx and elevates p-eEF2, leading to protein synthesis inhibition. mGluR stimulation in the presence of APOE4 activates the mTOR pathway, enhancing p-RPS6 without further altering p-eEF2 levels, and in turn rescues the protein synthesis inhibition.

consisting of 96% Neurobasal A, 1 x B27, 2 mM glutamine, and 1% penicillin–streptomycin.

## Extracellular recordings and data analyses

The extracellular signals were recorded by MEA System (MEA2100-HS120; Multichannel Systems). Neuronal cultures on DIV15 were treated with ApoE4 (10 nM) with and without addition of DHPG (50 $\mu M$), after 15 mins of ApoE4 addition. To administer HFS to the MEA chips, two electrodes (labeled A and B) were selected and an electrical stimulus of a single bipolar pulse (100 $\mu s$ at − 10 $\mu A$ followed by 100 $\mu s$ at + 10 $\mu A$) was alternatively applied after 10 ms. For HFS, 20 trains of paired stimulation (100 $\mu s$ at − 10 $\mu A$ followed by 100 $\mu s$ at + 10 $\mu A$) at 100 Hz were applied 20 times at 4-s intervals (Fig S1C). Electrical recordings were taken before and after the HFS

**Figure 4. mGluR-mediated activation of the mTOR pathway in the presence of APOE4 increases phosphorylation of RPS6 without altering p-eEF2 levels.**
**(A)** Schematic showing the workflow of the experiment. DIV15 rat primary cortical neurons were treated with 10 nM APOE4 recombinant protein for 20 min. The neurons were stimulated with 50 $\mu M$ DHPG in the last 5 min in the presence or absence of rapamycin (500 nM) followed by immunostaining for p-RPS6 and p-eEF2. **(B)** Representative images of DIV15 primary cortical neurons indicating Tuj1 (top) and p-RPS6 intensities (middle) on APOE4 treatment followed by DHPG stimulation with or without rapamycin. The Tuj1 channel was used to select the region of interest (ROI) of the neuron (bottom) for analysis. Scale bar—15 $\mu m$. **(C)** Quantification of the p-RPS6 intensity in the whole neuron normalized to Tuj1 intensity in the whole neuron (ROI of the neuron indicated) on APOE4 treatment followed by DHPG stimulation with or without rapamycin. N = 29–36 neurons from three independent experiments, one-way ANOVA ($P < 0.0001$) followed by Tukey's multiple comparison test. **(D)** Quantification of the p-RPS6 intensity in dendrites normalized to Tuj1 intensity in dendrites (area marked by a dotted line in the ROI of the neuron was excluded for dendrite analysis) on APOE4 treatment with or without DHPG. N = 29–36 neurons from three independent experiments, one-way ANOVA ($P < 0.0001$) followed by Tukey's multiple comparison test. **(E)** Representative images of DIV15 primary cortical neurons indicating Tuj1 (top) and p-RPS6 intensities (middle) on DHPG treatment with or without rapamycin. The Tuj1 channel was used to select the ROI of the neuron (bottom) for analysis. Scale bar—15 $\mu m$. **(F)** Quantification of the FUNCAT intensity in the whole neuron normalized to Tuj1 intensity in the whole neuron (ROI of the neuron indicated) on DHPG treatment with or without rapamycin. N = 26–29 neurons from three independent experiments, one-way ANOVA ($P < 0.0001$) followed by Tukey's multiple comparison test. **(G)** Representative images of DIV15 primary cortical neurons indicating Tuj1 (top) and p-eEF2 intensities (middle) on APOE4 treatment followed by DHPG stimulation with or without rapamycin. The Tuj1 channel was used to select the ROI of the neuron (bottom) for analysis. Scale bar—15 $\mu m$. **(H)** Quantification of the p-eEF2 intensity in the whole neuron normalized to Tuj1 intensity in the whole neuron (ROI of the neuron indicated) on APOE4 treatment followed by DHPG stimulation with or without rapamycin. N = 25–32 neurons from three independent experiments, one-way ANOVA ($P < 0.0001$) followed by Tukey's multiple comparison test. **(I)** Quantification of the p-eEF2 intensity in dendrites normalized to Tuj1 intensity in dendrites (area marked by a dotted line in the ROI of the neuron was excluded for dendrite analysis) on APOE4 treatment followed by DHPG stimulation with or without rapamycin. N = 25–32 neurons from three independent experiments, one-way ANOVA ($P < 0.0001$) followed by Tukey's multiple comparison test. Source data are available for this figure.

**Table 1.  Antibodies used for immunostaining.**

| Protein | Dilution used for immunostaining | Catalog number, Company |
|---------|----------------------------------|-------------------------|
| Tuj1 | 1:1,000 | T8578; Sigma-Aldrich |
| p-eEF2 | 1:500 | 2331S; Cell Signaling Technologies |
| p-RPS6 | 1:500 | 2211S; Cell Signaling Technologies |
| Alexa Fluor 488 | 1:500 | A-11059; Thermo Fisher Scientific |
| Alexa Fluor 647 | 1:500 | A-21245; Thermo Fisher Scientific |

stimulation. The data acquisition and analyses were carried out using Multi Channel Suite (Multichannel Systems). The data were acquired at a sampling rate of 25 kHz, using the Butterworth second-order 300- to 3,000-Hz filter. Spikes were detected when the extracellularly recorded signals exceeded a threshold level set at ±5 $\sigma$, where $\sigma$ is the SD of the baseline noise during quiescent periods. The analyses of spikes, including plotting of raster plots, were done using MEAnalyzer software (Dastgheyb et al, 2020).

### Immunostaining for p-eEF2 and p-RPS6

DIV15 neurons were treated with APOE4 (10 nM) for 20 min, with DHPG (50 $\mu$M) added during the last 5 min. For experiments involving rapamycin, neurons were cotreated with DHPG and rapamycin (500 nM) simultaneously. After treatment, neurons were fixed with 4% PFA for 10 min, then washed three times with 1X PBS. Permeabilization was performed using 0.3% Triton X-100 in TBS50 for 10 min, followed by 1 h of incubation in blocking buffer (2% BSA and 2% FBS in TBS50T [0.1% Triton X-100]). Neurons were incubated overnight at 4°C with primary antibodies prepared in the blocking buffer (antibody dilutions and catalog numbers are listed in Table 1). After three washes with TBS50T, secondary antibody (detailed in Table 1) incubation was performed for 1 h at RT. Neurons were then washed three times with TBS50T and mounted on glass slides using Mowiol.

Imaging was performed using an Olympus FV3000 confocal laser scanning upright microscope with a 60X objective. The pinhole was set to 1 Airy Unit, and an optical zoom of 2X was applied to satisfy Nyquist's sampling criteria in the XY direction. Z-stacks were acquired with a 1 $\mu$m step size (~10–12 Z-slices) to capture planes above and below the focal plane. Image analysis was performed using FIJI. The Tuj1 channel was used to define neuronal ROIs. For dendritic measurements, the cell body intensity was blocked and measurements were taken from the remainder of the previously selected ROI. Maximum intensity projection of all Z planes was used to quantify mean fluorescence intensity. Fluorescence intensities of the p-eEF2 or p-RPS6 channels were normalized to the Tuj1 channel for comparison across treatment conditions.

### SDS–PAGE and Western blotting

The SDS–PAGE and Western blotting were performed as mentioned in our previous work (Ramakrishna et al, 2024). Denatured lysates were resolved on 10% gels and transferred overnight to PVDF membranes. Membranes were blocked with 5% BSA in TBST (TBS with 0.1% Tween-20) and incubated with primary antibodies (2–3 h, RT), followed by incubation with HRP-conjugated secondary antibodies (1 h, RT) (Table 2). After TBST washes, signals were detected by chemiluminescence. For eEF2 and RPS6 phosphorylation analysis, duplicate gels were run: one for phosphoproteins (p-eEF2, p-RPS6) and the other for total proteins (eEF2, RPS6), with Tuj1 as the loading control for both. Antibodies used against p-eEF2, p-RPS6, and Tuj1 were the same in both immunostaining and Western blotting. Blots were sectioned to probe paired proteins on the same membrane, and quantification was done by densitometric analysis in ImageJ.

### RNA isolation and qRT–PCR

DIV15 rat cortical neurons were treated with APOE4 recombinant protein for 20 min after which RNA was isolated from the cells using the standard TRIzol RNA extraction method (Cat No. 15596018; Thermo Fisher Scientific). The isolated RNA was reverse-transcribed to cDNA using random hexamers (Cat No. N8080127; Thermo Fisher Scientific) and MMLV (2680A; PrimeScript Reverse Transcriptase) followed by quantitative PCR. qRT–PCR data were analyzed by the absolute quantification method using a standard curve. Absolute copy numbers for *Camk2α*, *Homer1*, and *α-Tubulin* mRNA were obtained using primers specific to each of these

**Table 2.  Antibodies used for immunoblotting.**

| Protein | Dilution used for immunoblotting | Catalog number, Company |
|---------|----------------------------------|-------------------------|
| Tuj1 | 1:10,000 | T8578; Sigma-Aldrich |
| p-eEF2 | 1:1,000 | 2331S; Cell Signaling Technologies |
| eEF2 | 1:1,000 | 2332S; Cell Signaling Technologies |
| p-RPS6 | 1:1,000 | 2211S; Cell Signaling Technologies |
| RPS6 | 1:1,000 | 2217S; Cell Signaling Technologies |
| PABP1 | 1:1,000 | 4992S; Cell Signaling Technologies |
| Secondary rabbit HRP | 1:10,000 | A0545; Sigma-Aldrich |
| Secondary mouse HRP | 1:10,000 | 31430; Thermo Fisher Scientific |

**Table 3.  Primer details for qPCR.**

| mRNA | Forward primer (5′-3′) | Reverse primer (3′-5′) |
|---|---|---|
| Camk2α | AGACACCAAAGTGCGCAAAC | GGGTCGCACATCTTCGTGTA |
| Homer1 | GAAGCTGTCTTTTGCGTTTTTGT | TGCCCAATGAAAACTTGCTGT |
| α-Tubulin | TATGCCAAGCGTGCCTTTGT | TGAAAGCAGCACCTTGTGAG |

transcripts. The copy numbers of these mRNAs were normalized to the untreated condition (Table 3).

## Statistical analyses

All statistical analyses were performed using GraphPad Prism software. The normality of the data was checked using the Kolmogorov–Smirnov test. FUNCAT and immunostaining data were represented as violin plots. One-way ANOVA was used for multiple group comparisons, followed by Tukey's multiple comparison test. Only for the MEA result analysis, repeated-measures two-way ANOVA was used followed by Sidak's multiple comparison test. For comparisons between two conditions, an unpaired $t$ test was used. For MEA results, a paired $t$ test was used to compare between the two conditions. A $P$-value of less than 0.05 was considered statistically significant.

# Data Availability

The authors confirm that the data supporting the findings of this study are available as individual source data files accompanying this article.

# Supplementary Information

# Acknowledgements

We thank Prof. Deepak Nair from the Centre for Neuroscience, Indian Institute of Science, India, for providing rapamycin. This work was primarily supported by the core funds from the Centre for Brain Research (CBR) and the CBR-FABRIC grant. RS Muddashetty thanks the funding from CBR. BK Radhakrishna received a research fellowship from University Grants Commission (NTA Ref number: 191620053012), Government of India. V Gautam would like to acknowledge funding support from Indian Council of Medical Research (ICMR) and Anusandhan National Research Foundation (ANRF), India. A Chakraborty was supported by a fellowship from DBT Junior Research Fellowship (DBT-JRF) Program (Ref- DBT/2023-24/IISc/2202), Department of Biotechnology, Government of India. We thank all the central facilities at CBR-IISc, especially the CAF (Central Animal Facility) at IISc, Liquid Nitrogen Facility at IISc, Imaging Facility at CBR, and Primary Cell Culture Facility at CBR.

## Author Contributions

BK Radhakrishna: conceptualization, data curation, formal analysis, validation, investigation, methodology, project administration, and writing—original draft, review, and editing.

AP Kaladiyil: conceptualization, data curation, formal analysis, validation, investigation, methodology, project administration, and writing—original draft, review, and editing.

A Chakraborty: data curation, formal analysis, validation, investigation, visualization, methodology, and writing—review and editing.

V Gautam: resources, supervision, investigation, visualization, methodology, and writing—review and editing.

RS Muddashetty: conceptualization, resources, data curation, supervision, funding acquisition, investigation, project administration, and writing—review and editing.

## Conflict of Interest Statement

The authors declare that they have no conflict of interest.

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
