## [Reviewer comments · Life Science Alliance]

Group 1 mGluR Stimulation Rescues APOE4-Mediated Translation Defect in Neurons

Bindushree Radhakrishna, Ahamed Kaladiyil, Anushree Chakraborty, Vini Gautam, and Ravi Muddashetty
DOI: <https://doi.org/10.26508/lsa.202503287>

Corresponding author(s): Ravi Muddashetty, Centre for Brain Research, IISc (Bengaluru, India)

Review Timeline:	Submission Date:	2025-03-02
	Editorial Decision:	2025-04-09
	Revision Received:	2025-09-17
	Editorial Decision:	2025-10-13
	Revision Received:	2025-10-28
	Accepted:	2025-11-02

Scientific Editor: Sarita Hebbar

Transaction Report:

April 9, 2025

Re: Life Science Alliance manuscript #LSA-2025-03287-T

Dr. Ravi S Muddashetty
Indian Institute of Science Bangalore
Centre for Brain Research
CV Raman Avenue
Bengaluru, Karnataka 560012
India

Dear Dr. Muddashetty,

Thank you for submitting your manuscript entitled "Group 1 mGluR Stimulation Rescues APOE4-Mediated Translation Defect in Neurons" to Life Science Alliance. The manuscript was assessed by two expert reviewers, whose comments are appended to this letter.

Overall, the reviewers concluded that this topic is of significance to better understand Alzheimer's Disease development in ApoE4 carriers. However they raised concerns over the evidence in support of central claims made in this work, that need to be addressed (listed below) for publication here:

1. We agree with Reviewer 1 that additional data is needed to strengthen these observations, namely:
 - (a) showing evidence of altered NMDAR-mediated calcium signalling following suggestions in point 1.
 - (b) including the missing controls in Figure 3 (point 5)
2. Reviewers note that the manuscript relies heavily on FUNCAT/immuno-related staining to draw conclusions (Reviewer 1 point 3, Reviewer 2 point 1).
In line with this, we suggest the authors rule out that ApoE4 influences mRNA instability with your method of choice as indicated by Reviewer 1, point 3.
3. We concur with Reviewer 2 that authors should revise the discussion and better highlight the significance of their results (point 3). Likewise, Reviewer 1 also notes that in the absence of direct evidence in this manuscript, the authors should tone down their conclusions on the link between ApoE4 and mGluR signalling (point 6).

Given the overall recommendations, we would like to invite you to submit a revised manuscript addressing the Reviewers' comments. Please include a letter addressing all the reviewers' comments point by point.

Thank you for this interesting contribution to Life Science Alliance. We are looking forward to receiving your revised manuscript.

Sincerely,

Sarita Hebbar, PhD
Scientific Editor
Life Science Alliance

B. MANUSCRIPT ORGANIZATION AND FORMATTING:

Reviewer #1 (Comments to the Authors (Required)):

This manuscript focuses on APOE4-associated neuropathologic mechanisms and suggests an avenue by which these effects could be countered. Understanding the role of APOE4 is of major significance to neurodegeneration researchers, as human ApoE4 carriers are significantly more likely to develop late-onset Alzheimer's Disease.

The authors demonstrate that the addition of recombinant human APOE4 protein to primary cortical neurons leads to impaired translation, as indicated by FUNCAT quantification. There is a clear 'rescue' of translation following activation of mGluR1/5 using agonist DHPG. This rescue is independent of NMDAR (calcium-mediated) translation inhibition induced by APOE4, through the mTOR pathway; therefore, there is an NMDAR-independent reinstatement of translation (mGluR PI3K AKT mTOR S6K pS6) in the presence of APOE4. The neurons appear to be in comparably good health across all conditions and the treatment conditions follow a logical progression through Figures 1 to 3. Despite these positives, I regretfully submit to you that there are many areas of concern, which I will outline in the numbered comments below:

1. The authors claim that recombinant APOE4 inhibits translation in rat primary neuron culture by impairing NMDAR-mediated, calcium-dependent translation. Yet, an initial characterization of APOE4-induced effects is missing, especially based on the claims made about how APOE4 affects neurons. For instance: (i) there is no evidence of NMDAR-APOE4 colocalization; (ii) no quantification of changes in free cytosolic Ca²⁺ with a calcium dye or genetic calcium influx indicator (e.g. with GCaMP); and somewhat related to these, (iii) no recombinant protein control (effects could be due simply to the presence of a foreign extracellular human protein in a murine cell culture). Essentially, what's missing is evidence of NMDAR-mediated calcium signaling aberration.
2. The authors show that translation is restored by stimulating mGluR1/5 with DHPG, via phosphorylation of RPS6. Yet, Figure 2C shows that the addition of APOE4 does not reduce p-RPS6, and in fact leads to a trending (non-significant) increase in p-RPS6 relative to untreated baseline. Both APOE4+ conditions in Figure 2C lead to increased p-RPS6.
3. Since FUNCAT intensity is clearly reduced in the presence of APOE4, but p-RPS6 seems to be less affected, could APOE4 be affecting mRNA stability instead? This could explain reduced translation despite the elevated p-RPS6. Some staining of mRNA stabilizing proteins and a quantification of mRNA levels could help to develop a stronger link between the addition of APOE4 and inhibition of translation.
4. There is an inconsistent method of quantification when moving from Figure 1 to Figures 2 & 3, whereby effects in neurites (dendrites + axons) are quantified only in Figure 1.
5. Figure 3 does not show the effects of all control conditions. There should be a clear demonstration of the effects on FUNCAT

of (i) APOE4+, RAPA+; (ii) RAPA+; (iii) DHPG+; (iv) DHPG+, RAPA+.

6. Line 128 states "This is the first report that studies the effects of APOE4 on mGluR response". I would contend that this evidence shows that mGluR activation can counter the translation inhibition effects induced by APOE4 treatment, but not that there is a link between APOE4 and mGluR signaling.

7. While I can acknowledge the mention of future studies needing to focus on synaptic plasticity (lines 198-203), I believe that any mention of synaptic plasticity requires at least some directly associated read-out. These could be (i) LTP markers (phosphorylated receptor subunits, surface receptor quantification), (ii) Dendritic spine measures (morphology, total spine counts).

I think that this manuscript could be strengthened by addressing the points outlined above.

Reviewer #2 (Comments to the Authors (Required)):

Review:

Based in the knowledge that APOE4 impacts synaptic plasticity and neuronal function, this study investigates the potential activation of metabotropic glutamate receptors (mGluRs) in mitigating synaptic deficits. The authors demonstrate that stimulation of group I mGluRs rescues the APOE4-mediated reduction in neuronal de novo protein synthesis, a process driven by enhanced phosphorylation of Ribosomal Protein S6 (RPS6) downstream of the mTOR signaling pathway. These findings provide valuable insights into the molecular mechanisms underlying synaptic dysfunction associated with APOE4 and highlight the therapeutic potential of mGluR modulation, however the results obtained are not strong enough to confidently support this statement. Notably, this is the first study to report mGluR responses in the context of APOE4, revealing a previously unexplored interaction that may represent a compensatory mechanism against APOE4-induced deficits. However, the current experiments rely on a simple in vitro model to measure protein synthesis, without direct functional assessments. To better establish the therapeutic relevance of targeting mGluRs, future studies should determine whether activating this pathway reverses APOE4-induced neuronal deficits in functionally relevant models.

I have provided the following comments that, if adequately addressed, will enhance the quality of the manuscript.

1. It is necessary to include a functional experiment to demonstrate that the effect is real, because measuring protein synthesis in individual neurons and then averaging and comparing each neuron does not seem entirely accurate. With a sample size of $n = 30$ neurons, small changes may appear significant, but without having a functional impact. If these changes do not lead to a functional impact, then the findings may lack meaningful significance.

2. The methods are repetitive, as the immuno-related section has already been explained in the FUNCAT (Fluorescent Non-Canonical Amino Acid Tagging) part. By presenting it this way, it becomes more evident that all the results are based on imaging and fluorescence signal quantification, which may introduce some bias.

3. The discussion is missing and needs to be included to properly contextualize the results within the existing literature and highlight their significance.

Reviewer's comments and rebuttals

Reviewer 1

This manuscript focuses on APOE4-associated neuropathologic mechanisms and suggests an avenue by which these effects could be countered. Understanding the role of APOE4 is of major significance to neurodegeneration researchers, as human ApoE4 carriers are significantly more likely to develop late-onset Alzheimer's Disease. The authors demonstrate that the addition of recombinant human APOE4 protein to primary cortical neurons leads to impaired translation, as indicated by FUNCAT quantification. There is a clear 'rescue' of translation following activation of mGluR1/5 using agonist DHPG. This rescue is independent of NMDAR (calcium-mediated) translation inhibition induced by APOE4, through the mTOR pathway; therefore, there is an NMDAR-independent reinstatement of translation (mGluRPI3KAKTmTORS6KpS6) in the presence of APOE4. The neurons appear to be in comparably good health across all conditions and the treatment conditions follow a logical progression through Figures 1 to 3. Despite these positives, I regretfully submit to you that there are many areas of concern, which I will outline in the numbered comments below:

We thank the reviewer for the appreciation of our work and for the critical comments. We have addressed the concerns raised to the best of our ability.

The list of the changes made in the figures and the text is included at the end of this document. The responses to the comments are as follows –

1. The authors claim that recombinant APOE4 inhibits translation in rat primary neuron culture by impairing NMDAR-mediated, calcium-dependent translation. Yet, an initial characterization of APOE4-induced effects is missing, especially based on the claims made about how APOE4 affects neurons. For instance: (i) there is no evidence of NMDAR-APOE4 colocalization; (ii) no quantification of changes in free cytosolic Ca²⁺ with a calcium dye or genetic calcium influx indicator (e.g. with GCaMP); and somewhat related to these, (iii) no recombinant protein control (effects could be due simply to the presence of a foreign extracellular human protein in a murine cell culture). Essentially, what's missing is evidence of NMDAR-mediated calcium signalling aberration.

We kindly refer the reviewer to an earlier study from our lab that extensively characterizes the effects of APOE4 on neurons (Ramakrishna et al., 2021). (i) Although, we have not shown it in our study, APOE4-NMDAR colocalization has been shown by a different group (Konings et al., 2021). In the earlier study from our lab (Ramakrishna et al., 2021), the authors have shown that (ii) APOE4 induces calcium influx into neuronal cytoplasm by activating the NMDA Receptors (Ramakrishna et al., 2021). (iii) In the APOE4-treated neurons, the authors found NMDA Receptors to be non-functional with respect to its protein synthesis response, thus identifying NMDAR-mediated calcium signaling aberration (Ramakrishna et al., 2021). It was also shown

earlier that the effects of large calcium influx and extended protein synthesis inhibition are not seen on addition of the APOE3 protein, the most common human isoform of APOE, in the same murine cell culture conditions (Figure 6, Figure 7, Ramakrishna et al., 2021). This acted as a control for the exogenous introduction of a human protein into a murine system. Usage of RAP, an antagonist of the APOE receptor, also inhibited any response to APOE4 addition confirming the effect is specific to APOE4 (Figure 2, Ramakrishna et al., 2021).

In the current study, we do not attempt to reproduce many of our earlier results to avoid redundancy. However, we do reproduce the key responses of APOE4, relevant in the context of protein synthesis, which include APOE4-mediated protein synthesis inhibition observed as decreased FUNCAT and increased p-eEF2 levels (Figure 1C,1D,2F,2G).

2. The authors show that translation is restored by stimulating mGluR1/5 with DHPG, via phosphorylation of RPS6. Yet, Figure 2C shows that the addition of APOE4 does not reduce p-RPS6, and in fact leads to a trending (non-significant) increase in p-RPS6 relative to untreated baseline. Both APOE4+ conditions in Figure 2C lead to increased p-RPS6.

We apologize for any lack of clarity in our explanations. We have previously shown that APOE4 induces a calcium-dependent increase in p-eEF2 levels, which then leads to a reduction of translation elongation, reducing protein synthesis (Ramakrishna et al., 2021). We reproduce this response in our current set of results as well, where 20 minutes of APOE4 exposure leads to an increase in p-eEF2 levels and reduced FUNCAT (Figure 1C,1D,2F,2G). However, we do not observe a significant alteration in p-RPS6 levels in APOE4 treated neurons as observed using immunocytochemistry and immunoblotting (Figure 2C, 2D). To strengthen this result, we have now used immunoblotting to measure the ratio of p-RPS6 to RPS6 to capture any functional changes in RPS6 phosphorylation (Figure 2D). Here again, we observe no change in phosphorylation of RPS6 on APOE4 treatment. Thus, S6 mediated translation regulation is unaffected by APOE4 and the translation inhibition is primarily due to calcium-p-eEF2 mediated pathway.

We propose a mechanism where DHPG (mGluR agonist) addition rescues the reduced translation not by recovering the p-eEF2 levels to basal but by independently increasing the phosphorylation of RPS6, which remains unaltered (non-significant change) by APOE4 treatment alone. The increase in p-RPS6 levels increases the rate of translation initiation, which increases global protein synthesis, rescuing FUNCAT levels.

3. Since FUNCAT intensity is clearly reduced in the presence of APOE4, but p-RPS6 seems to be less affected, could APOE4 be affecting mRNA stability instead? This could explain reduced translation despite the elevated p-RPS6. Some staining of mRNA stabilizing proteins and a quantification of mRNA levels could help to develop a stronger link between the addition of APOE4 and inhibition of translation.

We thank the reviewer for this interesting comment. As per the reviewer's suggestions, now we have examined for changes in the levels of an mRNA-stabilizing protein, PABP1. Immunoblot analysis of 20-minute APOE4-treated neuronal samples shows no significant changes in PABP levels (Supplementary Figure 1A). We further quantified the levels of a few representative mRNAs, which also showed no significant alterations on APOE4 treatment. We chose mRNAs coding for the following proteins: Homer1, a postsynaptic scaffolding protein, Camk2 α , a postsynaptic enzyme, α -tubulin, a cytoskeletal protein (Supplementary Figure 1B). Hence our results indicate the mRNA stability does not seem to be affected in neurons by APOE4 exposure. But a detailed and comprehensive analysis required for a firm conclusion on this subject is beyond the scope of this manuscript.

4. There is an inconsistent method of quantification when moving from Figure 1 to Figures 2 & 3, whereby effects in neurites (dendrites + axons) are quantified only in Figure 1.

We apologize for the inconsistency observed. We have now included dendritic FUNCAT intensity quantification in Figure 3D. The p-RPS6 and p-eEF2 responses in dendrites on mGluR stimulation with or without rapamycin in the background of APOE4 can be found in Figure 4D and 4I respectively.

5. Figure 3 does not show the effects of all control conditions. There should be a clear demonstration of the effects on FUNCAT of (i) APOE4+, RAPA+; (ii) RAPA+; (iii) DHPG+; (iv) DHPG+, RAPA+.

We thank the reviewer for the critical comment. We have included all the necessary control results in Figure 3.

Rapamycin treatment at the chosen concentration (500nM) and time period (5 minutes) was found to completely inhibit DHPG-induced increase in FUNCAT (Figure 3E, 3F) and p-RPS6 (Figure 4E, 4F)

As per the reviewer's suggestions, we have additionally measured the effects of RAPA and APOE4 + RAPA treatments on neuronal FUNCAT levels. We observed that 5 minutes of Rapamycin treatment reduced FUNCAT signal as expected and in line with other studies. Importantly, we also found that the addition of Rapamycin to APOE4-treated neurons did not affect APOE4's FUNCAT response (Figure 3G,3H).

6. Line 128 states "This is the first report that studies the effects of APOE4 on mGluR response". I would contend that this evidence shows that mGluR activation can counter the translation inhibition effects induced by APOE4 treatment, but not that there is a link between APOE4 and mGluR signaling.

We agree with the reviewer that APOE4 and mGluR signaling are not connected based on the evidence that we have provided. We have modified the statement to convey our conclusions better.

7. While I can acknowledge the mention of future studies needing to focus on synaptic plasticity (lines 198-203), I believe that any mention of synaptic plasticity requires at least some directly associated read-out. These could be (i) LTP markers (phosphorylated receptor subunits, surface receptor quantification), (ii) Dendritic spine measures (morphology, total spine counts). I think that this manuscript could be strengthened by addressing the points outlined above.

We thank the reviewer for their suggestion. We have addressed this here using Multi Electrode Array (MEA) as a measurement of overall neuronal/synaptic activity. We have chosen MEA as an assay that measures neuronal capacity to undergo synaptic potentiation High frequency stimulation (HFS) (Figure 1E). Using this assay, we observe that untreated neurons exhibited significant increase in the mean spike rate post HFS (Figure 1F). APOE4-treated neurons failed to respond to HFS (Figure 1G, 1H). However, mGluR stimulation in the background of APOE4 was found to recover this capacity, where we observed an increased spike activity on HFS stimulation (Figure 1G, 1H). This provides a strong evidence that mGluR stimulation can rescue APOE4-induced synaptic defects.

References

1. Ramakrishna, S., Jhaveri, V., Konings, S.C., Nawalpuri, B., Chakraborty, S., Holst, B., Schmid, B., Gouras, G.K., Freude, K.K. and Muddashetty, R.S., 2021. APOE4 affects basal and NMDAR-mediated protein synthesis in neurons by perturbing calcium homeostasis. *Journal of Neuroscience*, 41(42), pp.8686-8709.
2. Konings, Sabine C., Laura Torres-Garcia, Isak Martinsson, and Gunnar K. Gouras. "Astrocytic and neuronal apolipoprotein E isoforms differentially affect neuronal excitability." *Frontiers in Neuroscience* 15 (2021): 734001.

Reviewer #2 (Comments to the Authors (Required)):

Review:

Based in the knowledge that APOE4 impacts synaptic plasticity and neuronal function, this study investigates the potential activation of metabotropic glutamate receptors (mGluRs) in mitigating synaptic deficits. The authors demonstrate that stimulation of group I mGluRs rescues the APOE4-mediated reduction in neuronal de novo protein synthesis, a process driven by enhanced phosphorylation of Ribosomal Protein S6 (RPS6) downstream of the mTOR signaling pathway. These findings provide valuable insights into the molecular mechanisms underlying synaptic dysfunction associated with APOE4 and highlight the therapeutic potential of mGluR modulation, however the results obtained are not strong enough to confidently support this statement.

Notably, this is the first study to report mGluR responses in the context of APOE4, revealing a previously unexplored interaction that may represent a compensatory mechanism against APOE4-induced deficits. However, the current experiments rely on a simple in vitro model to measure protein synthesis, without direct functional assessments. To better establish the therapeutic relevance of targeting mGluRs, future studies should determine whether activating this pathway reverses APOE4-induced neuronal deficits in functionally relevant models.

I have provided the following comments that, if adequately addressed, will enhance the quality of the manuscript.

We sincerely thank the reviewer for acknowledging the novelty of the study and appreciating our work. We thank the reviewer for all the comments and suggestions. We have tried our best to address the concerns of the reviewer to make the manuscript suitable for publication. The list of the changes made in the figures and the text is included at the end of this document. The response to the comments is as follows –

1. It is necessary to include a functional experiment to demonstrate that the effect is real, because measuring protein synthesis in individual neurons and then averaging and comparing each neuron does not seem entirely accurate. With a sample size of $n = 30$ neurons, small changes may appear significant, but without having a functional impact. If these changes do not lead to a functional impact, then the findings may lack meaningful significance.

We agree with the reviewer that functional experiments are required to capture real effects on neuronal activity. In this regard, we have conducted further experiments using Multi Electrode Array to measure neuronal capacity to undergo synaptic potentiation High frequency stimulation (HFS). Using this assay, we observe that APOE4-treated neurons failed to respond to HFS. However, mGluR stimulation in the background of APOE4 was found to recover this capacity, where we observed an increased spike activity on HFS stimulation (Figure 1G, 1H). This provides stronger evidence that mGluR stimulation can rescue APOE4-induced synaptic defects (please also see our response to reviewer #1 comment 7).

2. The methods are repetitive, as the immuno-related section has already been explained in the FUNCAT (Fluorescent Non-Canonical Amino Acid Tagging) part. By presenting it this way, it becomes more evident that all the results are based on imaging and fluorescence signal quantification, which may introduce some bias.

We recognize that our use of imaging and fluorescence signal quantification as the only method for our conclusions may introduce bias. To increase confidence in our results, we have now added results from immunoblotting experiments where we measure p-RPS6/RPS6 levels and p-eEF2/eEF2 levels on APOE4 treatment and further mGluR stimulation (Figure 2D, 2G). The results from these experiments align with the results obtained using fluorescence imaging,

strengthening our hypothesis by eliminating bias. We have further added results from Multi Electrode Array supporting our hypothesis, introducing a new dimension in this study.

3. The discussion is missing and needs to be included to properly contextualize the results within the existing literature and highlight their significance.

We have added a separate discussion section as per the reviewer's suggestion. This helps us in signifying the importance of our results by aligning it with existing literature and drawing broader conclusions.

List of changes made in the manuscript

Figures

Figure number	Changes made
Figure 1E-H	MEA data added
Figure 2D, 2G	Immunoblotting data added
Figure 3	D: Dendritic analysis data added E-F: Moved from supplementary to main figure G-H: Controls data added
Figure 4	New figure B-C: moved from Figure 3 to 4 D: Dendritic analysis data added E-H: Moved from supplementary to main figure I: Dendritic analysis data added
Figure 5	New figure (Previously graphical abstract)
	New graphical abstract is added

Supplementary Figures

Figure number	Changes made
Figure S1	New figure

Main text

Separate discussion section is added.

October 13, 2025

RE: Life Science Alliance Manuscript #LSA-2025-03287-TR

Dr. Ravi S Muddashetty
Centre for Brain Research, IISc (Bengaluru, India)
Centre for Brain Research
CV Raman Avenue
C V Raman Avenue
Bengaluru, Karnataka 560012
India

Dear Dr. Muddashetty,

Thank you for submitting your revised manuscript entitled "Group 1 mGluR Stimulation Rescues APOE4-Mediated Translation Defect in Neurons". It was evaluated by one of the original reviewers whose comments are appended below. As you will read, the reviewer notes that the revised manuscript has addressed previous concerns. We agree with the reviewer and we would be happy to publish your paper in Life Science Alliance pending final revisions necessary to meet our formatting guidelines.

- Please complete the details for antibodies detecting phosphorylated and unphosphorylated proteins under both immunostaining and western blotting sections. If different antibodies were not used, please state that explicitly.
- We encourage you to include the source data for the manuscript or change the 'data availability' statement accordingly. As it is written now, it appears that all source data is provided. But neither is the information in a supplemental file nor is the link (Open Access Switchboard Metadata at REVISION) functional. You can alternatively indicate a future date when the link will be functional.
- Some images appear to be repeated in different figures. We ask you to check this. If the images are indeed the same, then it must be clearly stated in both figure legends that images are used in the other figure. Duplications are in:
Figure 2B and Figure S2B
Figure 2E and Figure S2D
- Please add your main, supplementary figure, and table legends to the main manuscript text after the references section.
- We encourage you to revise the figure legend for Figure 2 such that the figure panels are introduced in alphabetical order
- Please use the [10 author names, et al.] format in your references (i.e., limit the author names to the first 10)
- Please be sure that the authorship listing and order is correct

A. FINAL FILES:

-- Summary blurb (enter in submission system): A short text summarizing in a single sentence the study (max. 200 characters including spaces). This text is used in conjunction with the titles of papers, hence should be informative and complementary to

the title. It should describe the context and significance of the findings for a general readership; it should be written in the present tense and refer to the work in the third person. Author names should not be mentioned.

B. MANUSCRIPT ORGANIZATION AND FORMATTING:

Thank you for your attention to these final processing requirements. Please revise and format the manuscript and upload materials as soon as you are able.

Sincerely,

Sarita Hebbar, PhD
Scientific Editor
Life Science Alliance
<http://www.lsajournal.org>

Reviewer #1 (Comments to the Authors (Required)):

This manuscript focuses on APOE4-associated neuropathologic mechanisms and suggests an avenue by which these effects could be countered. Understanding the role of APOE4 is of major significance to neurodegeneration researchers, as human ApoE4 carriers are significantly more likely to develop late-onset Alzheimer's Disease.

The authors demonstrate that the addition of recombinant human APOE4 protein to primary cortical neurons leads to impaired translation, as indicated by FUNCAT quantification. There is a clear 'rescue' of translation following activation of mGluR1/5 using agonist DHPG. This rescue is independent of NMDAR (calcium-mediated) translation inhibition induced by APOE4, through the mTOR pathway; therefore, there is an NMDAR-independent reinstatement of translation (mGluR>PI3K>AKT>mTOR>S6K>pS6) in the presence of APOE4. The neurons appear to be in comparably good health across all conditions and the treatment conditions follow a logical progression through Figures 1 to 3.

My previous critiques and questions have been adequately addressed.

October 30, 2025

RE: Life Science Alliance Manuscript #LSA-2025-03287-TRR

Dr. Ravi S Muddashetty
Centre for Brain Research, IISc (Bengaluru, India)
Centre for Brain Research
CV Raman Avenue
C V Raman Avenue
Bengaluru, Karnataka 560012
India

Dear Dr. Muddashetty,

Thank you for submitting your Research Article entitled "Group 1 mGluR Stimulation Rescues APOE4-Mediated Translation Defect in Neurons". It is a pleasure to let you know that your manuscript is now accepted for publication in Life Science Alliance. Congratulations on this interesting work.

Your manuscript will now progress through copyediting and proofing. At the proofs stage, we suggest that you make a minor edit to the title, so that it reads as, "Group 1 mGluR Stimulation Rescues APOE4-Mediated Translation Defects in Neurons". It is journal policy that authors provide original data upon request.

DISTRIBUTION OF MATERIALS:

Again, congratulations on a very nice paper. I hope you found the review process to be constructive and are pleased with how the manuscript was handled editorially. We look forward to future exciting submissions from your lab.

Sincerely,

Sarita Hebbar, PhD
Scientific Editor
Life Science Alliance
<http://www.lsajournal.org>